# Effects of histamine on human periodontal ligament fibroblasts under simulated orthodontic pressure

Marcella Groeger[1], Gerrit Spanier[2], Michael Wolf[3], James Deschner[4], Peter Proff[1], Agnes Schröder[1☉], Christian Kirschneck[1☉]*

1 Department of Orthodontics, University Hospital Regensburg, Regensburg, Germany, 2 Department of Cranio-Maxillo-Facial Surgery, University Hospital Regensburg, Regensburg, Germany, 3 Department of Orthodontics, University Hospital RWTH Aachen, Aachen, Germany, 4 Department of Periodontology and Operative Dentistry, University of Mainz, Mainz, Germany

☉ These authors contributed equally to this work.
* christian.kirschneck@ukr.de

**Data Availability Statement:** All relevant data are within the paper and its Supporting Information files.

## Abstract

As type-I-allergies show an increasing prevalence in the general populace, orthodontic patients may also be affected by histamine release during treatment. Human periodontal ligament fibroblasts (PDLF) are regulators of orthodontic tooth movement. However, the impact of histamine on PDLF in this regard is unknown. Therefore PDLF were incubated without or with an orthodontic compressive force of 2g/cm$^2$ with and without additional histamine. To assess the role of histamine-1-receptor (H1R) H1R-antagonist cetirizine was used. Expression of histamine receptors and important mediators of orthodontic tooth movement were investigated. PDLF expressed histamine receptors H1R, H2R and H4R, but not H3R. Histamine increased the expression of H1R, H2R and H4R as well as of interleukin-6, cyclooxygenase-2, and prostaglandin-E2 secretion even without pressure application and induced receptor activator of NF-kB ligand (RANKL) protein expression with unchanged osteoprotegerin secretion. These effects were not observed in presence of H1R antagonist cetirizine. By expressing histamine receptors, PDLF seem to be able to respond to fluctuating histamine levels in the periodontal tissue. Increased histamine concentration was associated with enhanced expression of proinflammatory mediators and RANKL, suggesting an inductive effect of histamine on PDLF-mediated osteoclastogenesis and orthodontic tooth movement. Since cetirizine inhibited these effects, they seem to be mainly mediated via histamine receptor H1R.

## Introduction

In the dental specialty of orthodontics malpositioned teeth, which give rise to functional problems as well as affect facial aesthetics, are moved into their correct physiological position within the alveolar bone by fixed or removable orthodontic appliances via the application of mechanical forces in the direction of required movement [1]. These forces promote the

**Funding:** This study was supported by the Verein zur Förderung der wissenschaftlichen Zahnheilkunde in Bayern e.V. (C.K., grant nr. Kirschneck 2018, http://www.vfwz.de) and the German Orthodontic Society DGKFO (A.S., grant nr. Schröder 2019, https://www.dgkfo-vorstand.de/).

**Competing interests:** The authors have declared that no competing interests exist.

formation of tensile and pressure zones in the periodontal ligament, which connects teeth to their surrounding alveolar bone. As a result, a sterile inflammatory reaction occurs in the periodontal ligament, which is mainly mediated by periodontal ligament fibroblasts (PDLF) [2,3], but also involves cells of the immune system such as macrophages, lymphocytes and T cells [4,5]. Stimulation by mechanical forces induces secretion of proinflammatory enzymes, cytokines and chemokines by PDLF [3,6,7]. Furthermore PDLF enhance receptor activator of NF-kB ligand (RANKL) expression and reduce osteoprotegerin secretion upon pressure application [3,6,8], thus promoting differentiation of osteoclast progenitor cells to bone-resorptive osteoclasts [7,9]. According to the biphasic theory of orthodontic tooth movement (OTM), movement is achieved via mechanically stressed PDLF- and lymphocyte-regulated bone resorption processes with subsequent bone formation via osteoclast-stimulated osteoblasts with numerous cell-cell interactions [2,3,10]. Despite the importance of orthodontic treatment for patient health, many aspects of orthodontic therapy have so far been poorly understood.

Nutrition is reported to influence the oral microflora in that an oral health-optimized diet can reduce inflammatory processes associated with gingivitis and periodontitis [11]. An influence of diet-induced obesity on periodontal bone loss has also been demonstrated with adiposity and the applied fatty acid profile modulating bone metabolism [12,13]. Histamine, on the one hand, can be absorbed through food, but is also released in the body as part of innate immunity. In food, histamine is produced by the bacterial degradation of the amino acid histidine [14]. Biochemically histamine is a biogenic amine, just as tyramine, serotonin, dopamine, epinephrine, norepinephrine or octopamine. It is formed by elimination of carbon dioxide from the amino acid histidine and stored in particular in mast cells, basophilic granulocytes and nerve cells [15]. Histamine is a natural product that acts as a tissue hormone and neurotransmitter in the human or animal organism and is also widely found in plants and bacteria [16]. In humans, histamine plays a central role in allergic reactions and is involved in the immune system and in the defense against foreign substances [15]. In the human body, histamine is produced by mast cells and released after an immune reaction [15,17]. It can cause a drop in blood pressure and allergic reactions such as itching or redness. Dietary histamine may, under certain conditions, also lead to such reactions, including poisoning [18].

Four different histamine receptors are currently known: H1R, H2R, H3R and H4R [19,20]. They are distinguished by their function, structure, distribution, and their affinity to histamine [21–23]. Histamine can have pro-inflammatory and anti-inflammatory effects, which are mediated by different histamine receptor subtypes and cell types [23]. The receptors H1, H2 and H4 are particularly responsible for the effects of histamine in defense reactions such as mast cell activation, release of interleukins, recruitment of leukocytes, erythema, gastric acid secretion, vomiting and enlargement and increase of permeability of small blood vessels [20]. H1R is involved in allergy and inflammation and responsible for cell migration, vasodilatation and nociception [24,25]. H2R is known to modify vascular permeability [26]. H3R plays an important role in neuro-inflammatory diseases [22]. Like H1R, H4R is involved in allergy and inflammation and mediates mast cell activation [27] In addition to H1R, H3R mediates the neurotransmitter functions of histamine [22,23].

Histamine was reported to promote osteoclastogenesis directly through autocrine and paracrine action on osteoclast progenitor cells and indirectly by increasing the RANKL/OPG ratio in osteoblasts indicating specific roles of H1R and H2R [28]. Furthermore H1R seems to be expressed in PDLF playing a role in $Ca^{2+}$ signalling [29]. As PDLF play an important role in the regulation of orthodontic tooth movement due to their mechanically induced expression of mediators resulting in alveolar bone remodeling, a possible impact of histamine on these cells in the context of OTM is of clinical interest, as repercussions on OTM velocity and possible side effects such as dental root resorption and periodontal bone loss could be relevant in

patients with allergies. It is, however, currently still unknown, whether increased histamine concentrations have an impact on PDLF-mediated orthodontic tooth movement in pressure zones of of the periodontal ligament.

## Material and methods

### Isolation of periodontal ligament fibroblasts (PDLF)

We isolated primary periodontal ligament fibroblasts (PDLF) from periodontal tissue of the middle third of human wisdom teeth, which were free of decay and extracted at our dental facility for medical reasons such as retention or displacement. We performed all experiments in accordance with relevant guidelines and regulations. We obtained approval to collect and use PDLF from the ethics committee of the University of Regensburg, Germany (approval number 12-170-0150). Informed consent was obtained from all participants and/or their legal guardian/s. A pool of PDLF from six gender-mixed patients (aged 17–27 years) was used to maximize data generalisability. PDLF cells of each individual subjects included into the pool were tested for increased COX-2 and RANKL gene expression upon compressive force treatment (S1 Fig). Tissue samples were incubated in 6-well-plates at 37˚C, 5% $CO_2$, 100% $H_2O$ in full media (dulbecco's modified eagle medium DMEM–high glucose, D5671, Sigma Aldrich, Munich, Germany), 10% FBS (fetal bovine serum, P30-3306, PAN-Biotech, Aidenbach, Germany), 10% L-Glutamine (G7513, Sigma Aldrich, Munich, Germany), 1% ascorbic acid (A8960, Sigma Aldrich, Munich, Germany), 1% antibiotics/antimycotics (A5955, Sigma Aldrich, Munich, Germany) until proliferation of fibroblasts [30]. We characterized the cells by fibroblast-specific marker gene expression and morphology, as described before [30]. Until use, they were frozen in liquid nitrogen and 90% FBS, 10% DMSO (dimethyl sulfoxide, A994.1, Carl Roth, Karlsruhe, Germany).

### Experimental design of cell culture experiments

PDLF of 3rd to 5th passage were used for the experiments. A total of 70,000 PDLF in 2 ml DMEM per well were randomly seeded onto 6-well-plates, and preincubated for 24 h with or without addition of 100 μM of histamine (H7125, Sigma Aldrich, Munich, Germany) (50, 100 and 200 μM in receptor expression experiments). After that time PDLF were left untreated or a glass plate ($2g/cm^2$) was applied for another 48 h to simulate orthodontic pressure in the periodontal ligament according to an established and published in-vitro model [30,31] (Fig 1). To test for histamine receptor (HR) interaction, we additionally incubated PDLF with 100 μM of H1R antagonist cetirizine (C3618, Sigma Aldrich, Munich, Germany), H2R antagonist ranitidine (R101, Sigma Aldrich, Munich, Germany) or H4R antagonist JNJ7777120 (J3770, Sigma Aldrich, Munich, Germany), respectively, two hours prior to histamine application. The used antagonist concentrations were adopted from the concentration of a H1R antagonist previously used and published in experiments on nasal fibroblasts [32]. Then PDLF were preincubated for 24 h followed by pressure application for another 48 h as described above. We then analyzed gene expression (RT-qPCR) and protein expression (Western Blot, ELISA).

### Determination of cell number

We harvested PDLF with a cell scraper in 1 ml PBS and quantified cell number using a Beckman Coulter Counter Z2™ (Beckman Coulter, Krefeld, Germany) according to the manufacturer's instructions.

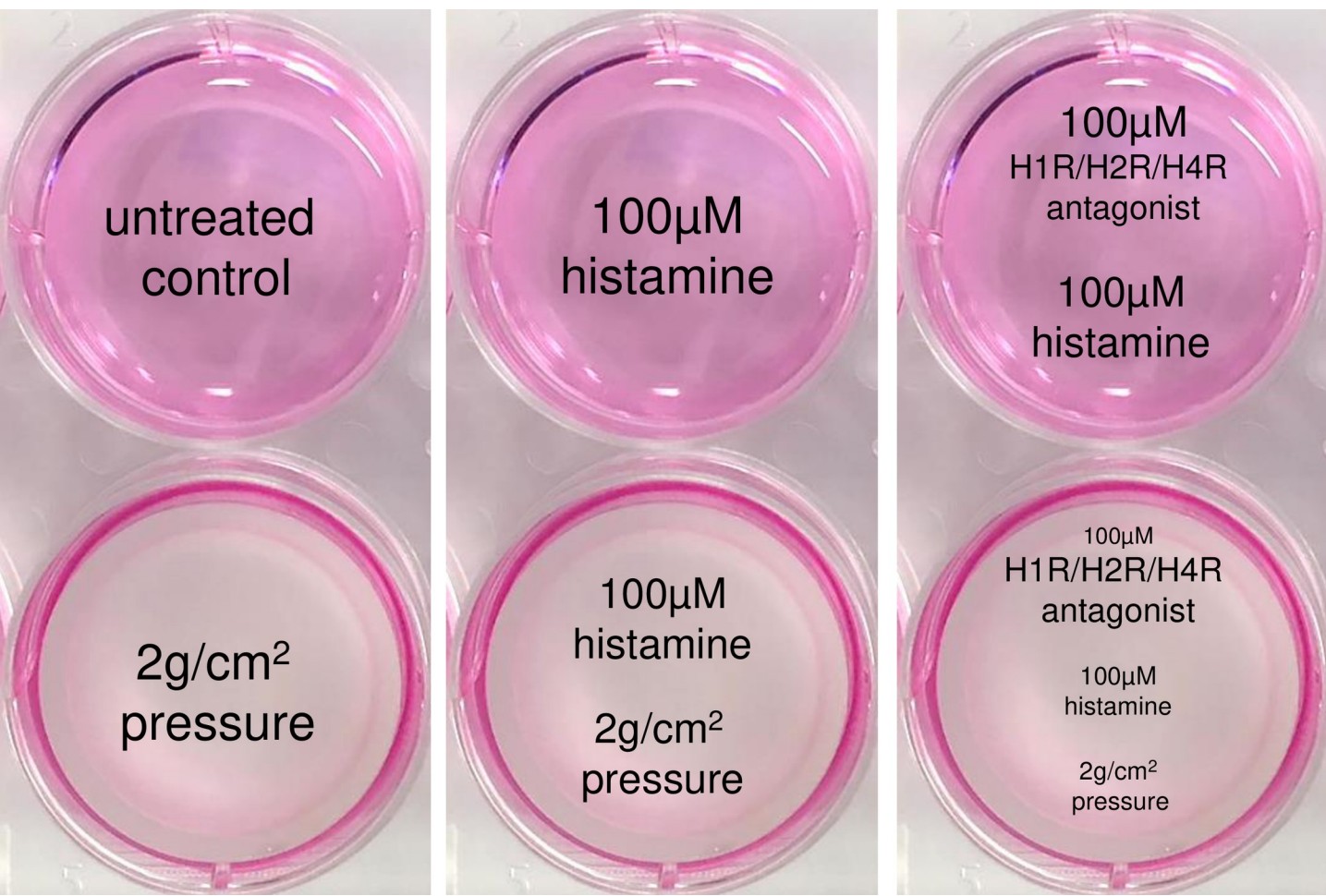

**Fig 1. *In vitro* simulation of compressive force application to periodontal ligament fibroblasts (PDLF) occurring during orthodontic tooth movement.** After a preincubation time of 24 h with or without histamine (100 μM, 50/100/200 μM in receptor expression experiments) and an H1R/H2R/H4R antagonist (100 μM), a pressure of 2g/cm² was applied to PDLF by means of a sterile glass disc (ø33cm, 17.1g) for 48 h according to an established and published *in vitro* model.

### Cytotoxicity assay (LDH release)

To determine cytotoxicity we used lactate dehydrogenase (LDH) assays (04744926001, Sigma Aldrich, Munich, Germany) following the manufacturer's instructions. Briefly, we added 100 μl of freshly prepared LDH solution containing of 22 μl catalyst mixed with 1 ml dye to 100 μl cell culture supernatant and incubated the mixture for 30 min in the dark at room temperature. We stopped the reaction by adding 50 μl stop solution. An ELISA reader (Multiscan GO Microplate Spectrophotometer, Thermo Fisher Scientific, Waltham, MA, USA) was used to measure LDH activity (absorbance at 490 nm), subtracting background absorbance at 690 nm.

### Isolation of total RNA

Total RNA from PDLF was isolated using 500 μl TriFast (peqGOLD, PEQLAB Biotechnology Erlangen, Germany) for each sample according to the manufacturer's instructions. The RNA pellet was eluted in 25 μl nuclease-free water (T143, Carl Roth, Karlsruhe, Germany) and RNA concentration was determined by measuring OD at 260 nm (NanoPhotometer, Implen, Munich, Germany).

## cDNA synthesis

For cDNA synthesis we mixed 1 μg of RNA with nuclease-free water to get a volume of 11 μl. This compound was applied to a mixture of 4 μl 5xM-MLV-buffer (M1705, Promega, Madison, WI, USA), 1 μl Oligo$_{dt}$ primer (SO131, Thermo Fisher Scientific, Waltham, MA, USA), 1 μl random hexamer primer (SO142, Thermo Fisher Scientific, Waltham, MA, USA), 1 μl 10 mM dNTP (L785.2, Carl Roth, Karlsruhe, Germany), 1 μl (40 U) RNase Inhibitor (EO0381, Thermo Fisher Scientific, Waltham, MA, USA) and 1μl (200 U) M-MLV Reverse Transcriptase (M1705, Promega, Madison, WI, USA) [6]. All samples were incubated at 37˚C for 1 h and at 95˚C for 2 min to inactivate the transcriptase. They were stored at -20˚C until use. To minimize experimental variations, all components were prepared as a master mix and cDNA synthesis was performed at the same for all samples.

## Semiquantitative PCR

We performed semiquantitative PCR and agarose gel electrophoresis to get information regarding histamine receptor expression in PDLF. For this purpose we mixed 2 μl of cDNA with 2 μl 10xFastStart PCR buffer with 20 mM MgCl$_2$ (12161567001, Sigma Aldrich, Munich, Germany), 0.5 μl of the appropriate forward and reverse primer respectively (Table 1), 0.4 μl dNTPs (L785.2, Carl Roth, Karlsruhe, Germany) and 0.2 μl FastStart Taq polymerase (12032929001, Sigma Aldrich, Munich, Germany) and added H$_2$O$_{dd}$ to a total volume of 20 μl. We used histamine receptor primer combinations according to the study of Park et al. [32] (Table 1). *RPL22* was used as reference gene, as it has been shown to be stably expressed before [30,33]. The samples were heated in a thermocycler (VWR, Radnor, PA, USA) at 95˚C for 5 minutes and went through 40 cycles at 60˚C for 30 seconds each. For agarose gel electrophoresis, we used a 1.5% agarose gel, which was prepared with agarose powder (T145.3, Carl Roth, Karlsruhe, Germany), 1xTris acetate EDTA buffer and gel red buffer (41003, Biotrend, Cologne, Germany). 7 μl of each sample were mixed with a 2 μl sucrose buffer and carefully pipetted into the pockets of the agarose gel. A voltage of 120 V was applied for 40 min in TAE buffer. The evaluation was then carried out using the gel documentation system Genoplex 2 and its software GenoSoft (VWR, Radnor, PA, USA). Densitometric analysis of specific bands was performed with ImageJ (ver. 1.47, Wayne Rasband, National Institutes of Health, USA).

## Quantitative real-time polymerase chain reaction (RT-qPCR)

We pipetted 7.5 μl SYBR®Green Jumpstart Taq ready mix (S4438, Sigma Aldrich, Munich, Germany), 5.25 μl nuclease-free water (T143, Carl Roth, Karlsruhe, Germany), 0.75 μl of a

**Table 1. Primer data for target genes and reference genes (*PPIB*, *RPL22*) for semiquantitative and RT-qPCR.**

| Gene symbol | Gene name | Accession Number | 5´-forward primer-3´ | 5´-reverse primer-3´ |
|---|---|---|---|---|
| *H1R* | histamine 1 receptor | NM_001098213.1 | GTCTAACACAGGCCTGGATT | GGATGAAGGCTGCCATGATA |
| *H2R* | histamine 2 receptor | NM_001131055.1 | ATTAGCTCCTGGAAGGCAGC | CTGGAGCTTCAGGGGTTTCT |
| *H3R* | histamine 3 receptor | NM_007232.3 | TCGTGCTCATCAGCTACGAC | AAGCCGTGATGAGGAAGTAC |
| *H4R* | histamine 4 receptor | NM_021624.4 | GGCTCACTACTGACTATCTG | CCTTCATCCTTCCAAGACTC |
| *COX2* | cyclooxygenase 2 | NM_000963.3 | GAGCAGGCAGATGAAATACCAGTC | TGTCACCATAGAGTGCTTCCAAC |
| *IL6* | interleukin 6 | NM_000600.3 | TGGCAGAAAACAACCTGAACC | CCTCAAACTCCAAAAGACCAGTG |
| *TNFRSF11B (OPG)* | osteoprotegerin | NM_002546.4 | TGTCTTTGGTCTCCTGCTAACTC | CCTGAAGAATGCCTCCTCACAC |
| *PPIB* | peptidylprolyl isomerase A | NM_000942.4 | TTCCATCGTGTAATCAAGGACTTC | GCTCACCGTAGATGCTCTTTC |
| *TNFSF11 (RANKL)* | receptor activator of NFκB ligand | NM_003701.3 | ATACCCTGATGAAAGGAGGA | GGGGCTCAATCTATATCTCG |
| *RPL22* | ribosomal protein L22 | NM_000983.3 | TGATTGCACCCACCCTGTAG | GGTTCCCAGCTTTTCCGTTC |

corresponding primer pair (0.375 µl / primer) and 1.5 µl cDNA, previously diluted to 1:10, per well onto a 96-well plate (712282, Biozym, Hessisch Oldendorf, Germany) in duplicates. To ensure equal concentrations, all components except the cDNA solution were prepared as a master mix. Amplification was performed with a Mastercycler ep realplex-S thermocycler (Eppendorf, Hamburg, Germany). At the beginning, the plate was heated to 95˚C for 5 min and went through 45 cycles with 95˚C each for 10 sec, 60˚C for 8 sec and 72˚C for 8 sec. At the end of each step fluorescence was quantified at 520 nm. $C_q$ values were determined using the software realplex (CalqPlex algorithm, automatic baseline). Normalization of target genes was based on two reference genes (*RPL22* and *PPIB*), which were validated before for PDLF and the used *in vitro* model [30,33]. We calculated relative gene expression as $2^{-\Delta C_q}$ [34] with $\Delta C_q$ = $C_q$ (target gene)–$C_q$ (mean *RPL22/PPIB*). All gene specific primers (Table 1) and gene nucleotide sequences were constructed according MIQE guidelines [35] using NCBI (National Centre for Biotechnology Information) PrimerBLAST and additional software considering final concentration of qPCR components [36]. Primers were synthesized by Eurofins MWG Operon LLC (Huntsville, AL, USA).

## Western Blot

Protein from PDLF was isolated using 100 µl CelLytic M (C2978; Sigma Aldrich, Munich, Germany) containing proteinase inhibitors (87786, Carl Roth, Karlsruhe, Germany) per well. We determined protein concentration with RotiQuant (K015.3; Carl Roth, Karlsruhe, Germany) according to the manufacturer's instructions. We separated equal amounts of total protein on a 10% SDS-polyacrylamide gel under reducing conditions and transferred the proteins onto a polyvinylidene diflouride (PVDF) membrane (T830, Carl Roth, Karlsruhe, Germany). Membranes were blocked with 5% nonfat milk in tris-buffered saline and 0.1% Tween 20, pH 7.5 (TBS-T) at 4˚C over night and incubated with anti-RANKL (TA306362, OriGene, Rockville, MD, USA) diluted 1:2,000 in 0.5% milk in TBS-T, or 1:500 anti-HSP90 (s-13119, Santa Cruz Biotechnology, Dallas, TX, USA) for 1 h. After washing three times in TBS-T, we incubated the blots for 1 h with anti-mouse IgG-κBP-HRP (sc-516102, Santa Cruz Biotechnology, Dallas, TX, USA) diluted 1:5,000 or anti-rabbit IgG HRP (611–1302, Rockland immunochemicals, Gilbertsville, PA, USA) diluted 1:5,000 in 5% milk in TBS-T horseradish peroxidase-conjugated anti-rabbit IgG (Thermo Fisher Scientific, Waltham, MA, USA), diluted 1:5000 in 0.5% milk in TBS-T at room temperature. After washing, antibody binding was visualized by the gel documentation system Genoplex 2 and its software (VWR, Radnor, PA, USA). Densitometric quantification of specific bands was performed with ImageJ (ver. 1.47, Wayne Rasband, National Institutes of Health, USA).

## Enzyme-linked immunosorbent assay (ELISA)

We used commercially available enzyme-linked immunosorbent assay (ELISA) kits (IL6: CSB-E04638h, Cusabio, Houston, TX, USA; osteoprotegerin (OPG): EHTNFRSF11B, Thermo Fisher Scientific, Waltham, MA, USA; RANKL: RD193004200R, BioVendor, Brno, Czech Republic) according to the manufacturers' instructions and measured absorbance using an ELISA plate reader (Multiskan Go, Thermo Fisher Scientific, Waltham, MA).

## Statistical methods

IBM SPSS Statistics 24 (IBM®, Armonk, NY, USA) was used for statistical analysis. Each symbol in figures represents a data point. Horizontal lines represent the mean ± standard error of mean. Data were validated by Welch-corrected ANOVAs with Games-Howell posthoc tests. All differences were considered statistically significant at $p \leq 0.05$.

## Results

### Expression of different histamine receptors (HR) and effects of HR-antagonists in PDLF

First, we focused on the expression levels of histamine receptors with variable concentrations of histamine. PDLF expressed histamine 1 receptor (*H1R*), histamine 2 receptor (*H2R*) and histamine 4 receptor (*H4R*, Fig 2A–2D). Histamine 3 receptor (*H3R*) was not expressed in PDLF (Fig 2A). Increasing histamine concentrations led to a significant increase of all expressed histamine receptors in PDLF using a concentration of 100 μM histamine (Fig 2A–2D). Application of 100 μM histamine increased *COX-2* gene expression significantly (Fig 2E). To determine which histamine receptor was responsible for this upregulation, we tested cetirizine which is a H1R antagonist, ranitidine as H2R antagonist and JNJ777210, which acts as H4R antagonist. We observed a significant reduction of *COX-2* gene expression after application of 100 μM histamine, when inhibiting H1R with cetirizine (Fig 2E). Neither ranitidine nor JNJ777210 seemed to inhibit histamine-induced *COX-2* upregulation at the mRNA level (Fig 2E).

### Effects of histamine and H1R antagonist cetirizine on cell number and cell viability

Next we investigated the effect of histamine on PDLF without and with compressive force treatment occurring during orthodontic tooth movement. Furthermore we investigated via which receptor histamine-induced effects are mediated. Histamine application increased PDLF number significantly without and with mechanical loading (Fig 3A). Inhibition of H1R with cetirizine truncated this effect (Fig 3A). Compressive force treatment reduced cell number significantly under all tested conditions (Fig 3A). In line with that, cytotoxicity was increased with pressure application under all tested conditions (Fig 3B), whereas LDH release was reduced after addition of histamine (Fig 3B). Again, treatment with cetirizine limited the histamine-induced effect (Fig 3B).

### Effects of histamine and H1R antagonist cetirizine on expression of proinflammatory genes in PDLF

Next, we investigated gene and protein expression of proinflammatory genes. Compressive force treatment increased *COX-2* gene expression significantly (Fig 4A). Stimulation of PDLF with 100 μM histamine led to an enhanced gene expression of *COX-2* under control conditions without pressure application (Fig 4A). Inhibition of H1R with cetirizine or fexofenadine (S2 Fig) reduced the histamine-induced *COX-2* expression to the control level without histamine (Fig 4A). We observed a pressure-induced upregulation of *COX-2* gene expression independent of histamine or cetirizine application (Fig 4A). In line with that PG-E2 secretion was enhanced after compression independent of treatment with histamine or cetirizine (Fig 4B). We observed increased PG-E2 secretion after treatment with histamine under control conditions and compressive force treatment, which was inhibited after adding cetirizine (Fig 4B). Under control conditions without pressure application histamine treatment increased *IL-6* gene expression and protein secretion significantly (Fig 4C and 4D). This effect could be inhibited by application of cetirizine (Fig 4C and 4D) or fexofenadine (S2 Fig). As expected, pressure application increased IL-6 gene and protein expression in PDLF (Fig 4C and 4D). Histamine application, however, truncated this pressure-induced IL-6 gene and protein expression (Fig 4C and 4D). Addition of cetirizine inhibited this histamine-induced effect at the mRNA and protein level (Fig 4C and 4D).

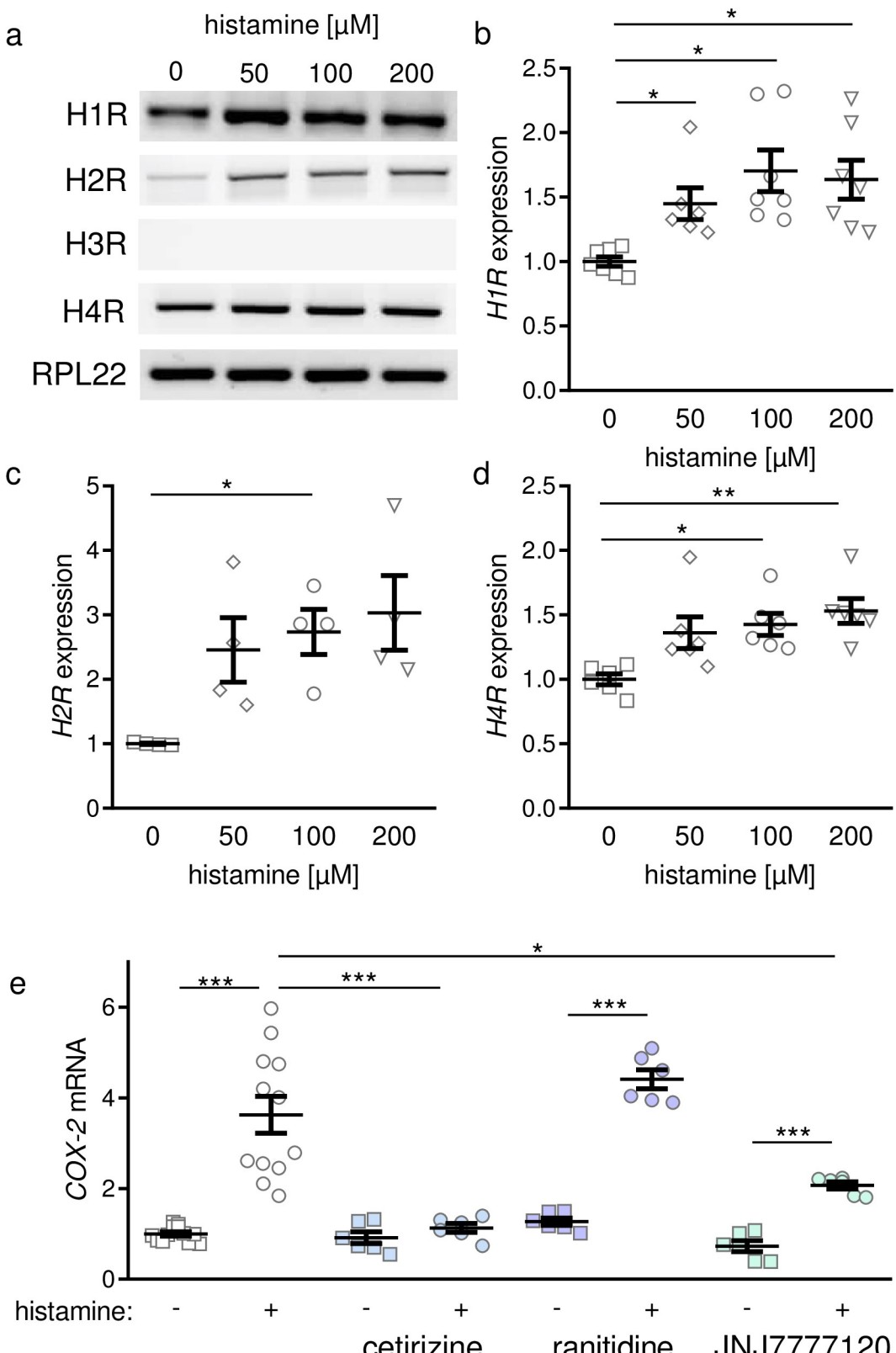

**Fig 2. Histamine-dependent expression of histamine receptors and COX-2 by periodontal ligament fibroblasts.** (a) Representative pictures of gene expression of histamine receptors in PDLF. Expression of histamine-3-receptor (*H3R*) could

not be determined. Densitometric analysis of semiquantitative PCR for histamine-1-receptor (*H1R*, b), histamine-2-receptor (H2R, c) or histamine-4-receptor (*H4R*, d) after treatment with different histamine concentrations. (e) *COX-2* gene expression after stimulation with histamine and inhibition of histamine receptor interaction using different inhibitors. AU: arbitrary units; $^*p \leq 0.05$; $^{**} p \leq 0.01$; $^{***} p \leq 0.001$. Statistics: Welch-corrected ANOVA with Games-Howell posthoc tests. Each symbol in figures represents a data point. Horizontal lines represent the mean ± standard error of mean.

### Effects of histamine and H1R antagonist cetirizine on the RANKL/OPG system in PDLF

Next we were interested in the impact of histamine on the PDLF-mediated remodeling of alveolar bone during simulated orthodontic tooth movement. Therefore we investigated the RANKL/OPG system. Compressive force treatment did not affect *OPG* (osteoprotegerin) mRNA expression in PDLF (Fig 5A). Histamine, however, increased *OPG* gene expression independent of pressure treatment. This effect was inhibited by cetirizine application (Fig 5A). In contrast to *OPG* gene expression we observed a decrease of secreted OPG protein in the media after pressure application (Fig 5B). Addition of histamine to cell culture media reduced OPG protein secretion with and without compression (Fig 5B). This reduction was counteracted by treatment with cetirizine. Pressure application resulted in significant *RANKL* gene expression under control conditions without histamine or cetirizine inhibition (Fig 5C). Histamine increased *RANKL* gene expression in PDLF without compression and reduced it to the control level after compressive force treatment (Fig 5C). Next we investigated RANKL secretion and protein expression of membrane-bound RANKL in PDLF. We observed enhanced RANKL secretion after mechanical loading in PDLF under control conditions (Fig 5D). Histamine treatment, however, reduced RANKL secretion without and with pressure application (Fig 5D). H1R inhibition via cetirizine administration restored this histamine-induced truncation of RANKL secretion without compression (Fig 5D). In line with soluble RANKL secretion, expression of membrane-bound RANKL on PDLF increased with pressure application (Fig 5E). Moreover, histamine enhanced membrane-bound RANKL expression with and without compressive force treatment (Fig 5E). Cetirizine application reversed this histamine effect on RANKL protein expression (Fig 5E). Next, we calculated RANKL/OPG mRNA ratio to directly assess the changes due to histamine or cetirizine treatment (Fig 5F). Under control conditions compression resulted in an increased RANKL/OPG mRNA ratio. Without pressure histamine elevated *RANKL/OPG* mRNA ratio significantly mediated by the H1R, as this effect was truncated by cetirizine (Fig 5F).

## Discussion

In this study we investigated the effect of histamine and histamine 1 receptor antagonist cetirizine on PDLF. Cetirizine has been shown to have an exquisite anti-H1R specificity exerting its effects only on H1R [37] and to facilitate bone formation by suppressing osteoclastic activity [38]. We determined that histamine receptors 1 (*H1R*), 2 (*H2R*) and 4 (*H4R*) are expressed by PDLF, whereas type 3 (*H3R*) is not. Gene expression of *H1R* and *H2R* significantly increased with histamine treatment. Furthermore histamine enhanced expression of interleukin-6 (*IL-6*), cyclooxygenase-2 (*COX-2*) and the secretion of prostaglandin E2 (PG-E2) by PDLF even without compression. RANKL protein expression was also induced, whereas OPG secretion remaining unaffected. Histamine significantly increased cell number and reduced LDH release. All mentioned effects were not observed during simultaneous incubation with the H1R antagonist cetirizine indicating that histamine effects were transmitted through H1R. Despite some interindividual variation of PDLF characteristics and expression behaviour, our results derived from a pool of PDLF from six gender-matched patients should be generalisable,

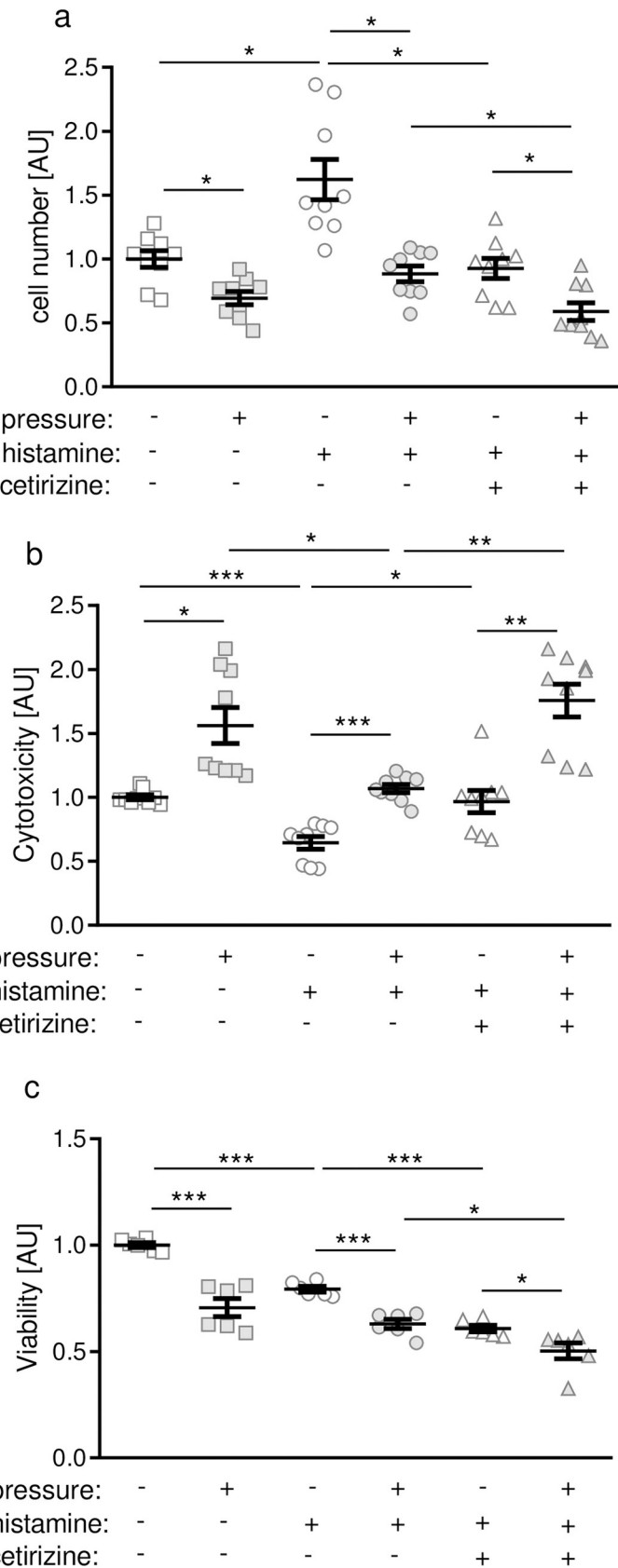

**Fig 3.** Assessment of cell number (a), cytotoxicity (b) and viability (c) after compression with or without histamine or inhibition with cetirizine. AU: arbitrary units; $^*$p $\leq$ 0.05; $^{**}$ p $\leq$ 0.01; $^{***}$ p $\leq$ 0.001. Statistics: Welch-corrected ANOVA with Games-Howell posthoc tests. Each symbol in figures represents a data point. Horizontal lines represent the mean ± standard error of mean.

as comparable upregulating effects by compressive force treatment were observed in all individual cell lines.

Our experiments indicate that PDLF only express *H1R*, *H2R* and *H4R* and not *H3R*. The reason could be that *H3R* is normally expressed by neurons and thus rather involved in neuro

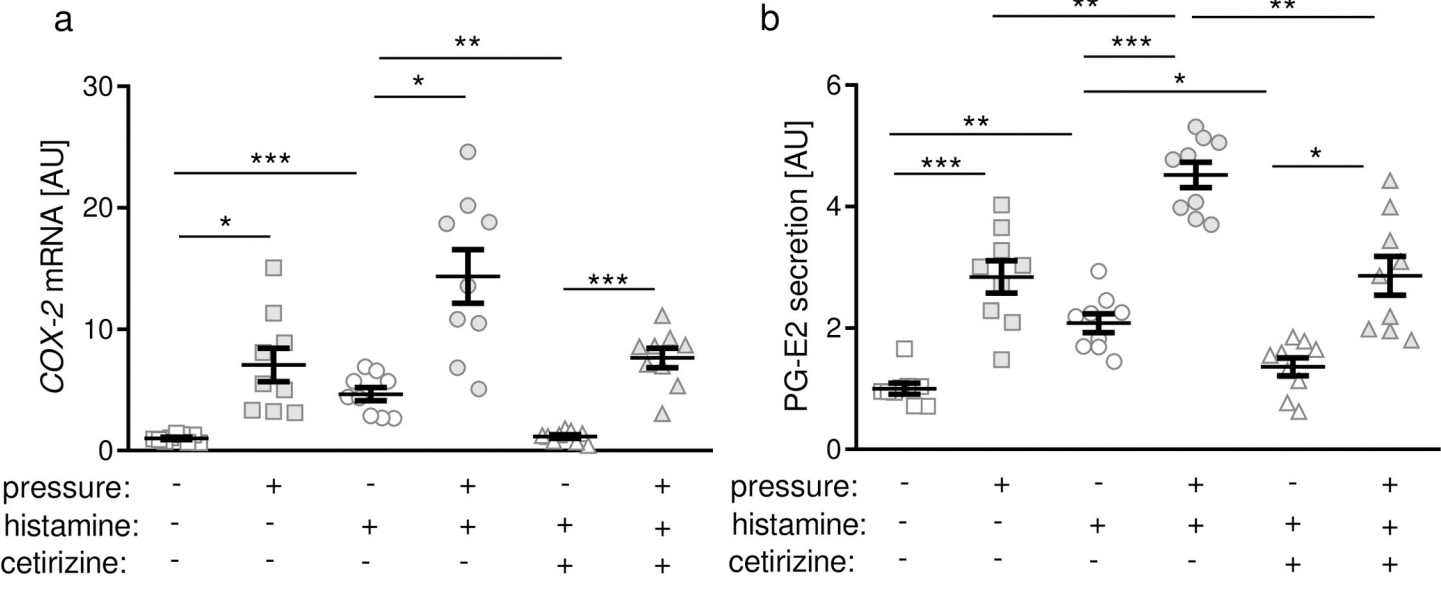

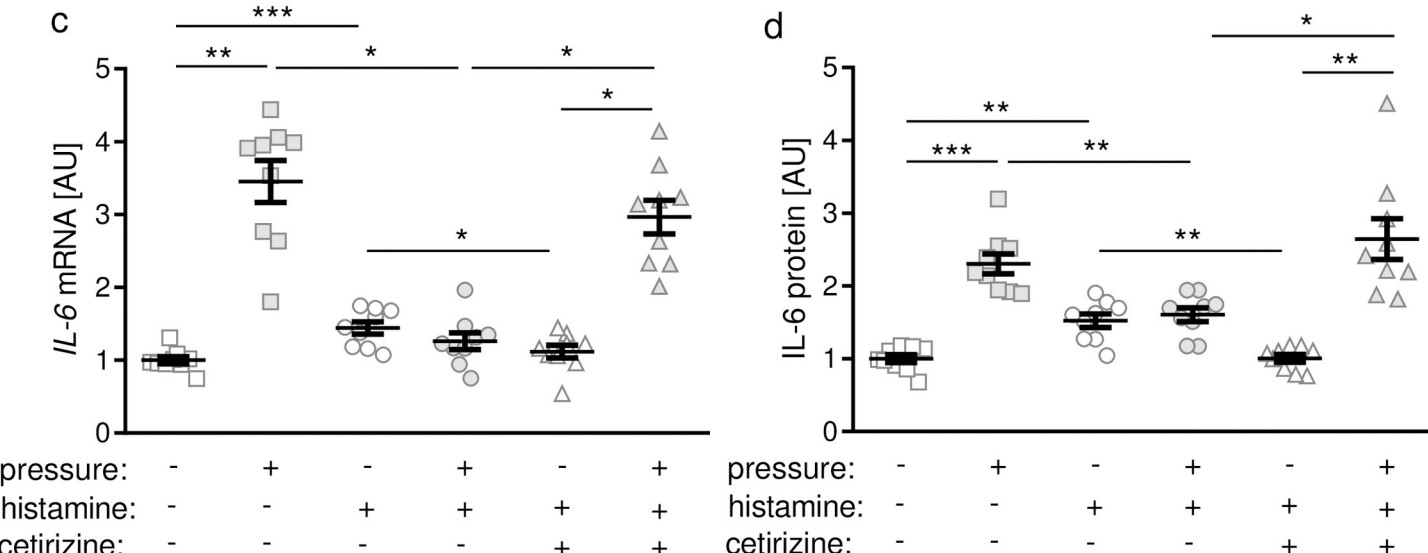

**Fig 4.** *COX-2* gene expression (a), PG-E2 secretion (b), *IL-6* gene expression (c) and IL-6 secretion (d) after compression with or without histamine or inhibition with cetirizine. AU: arbitrary units; $^*$p $\leq$ 0.05; $^{**}$ p $\leq$ 0.01; $^{***}$ p $\leq$ 0.001. Statistics: Welch-corrected ANOVA with Games-Howell posthoc tests. Each symbol in figures represents a data point. Horizontal lines represent the mean ± standard error of mean.

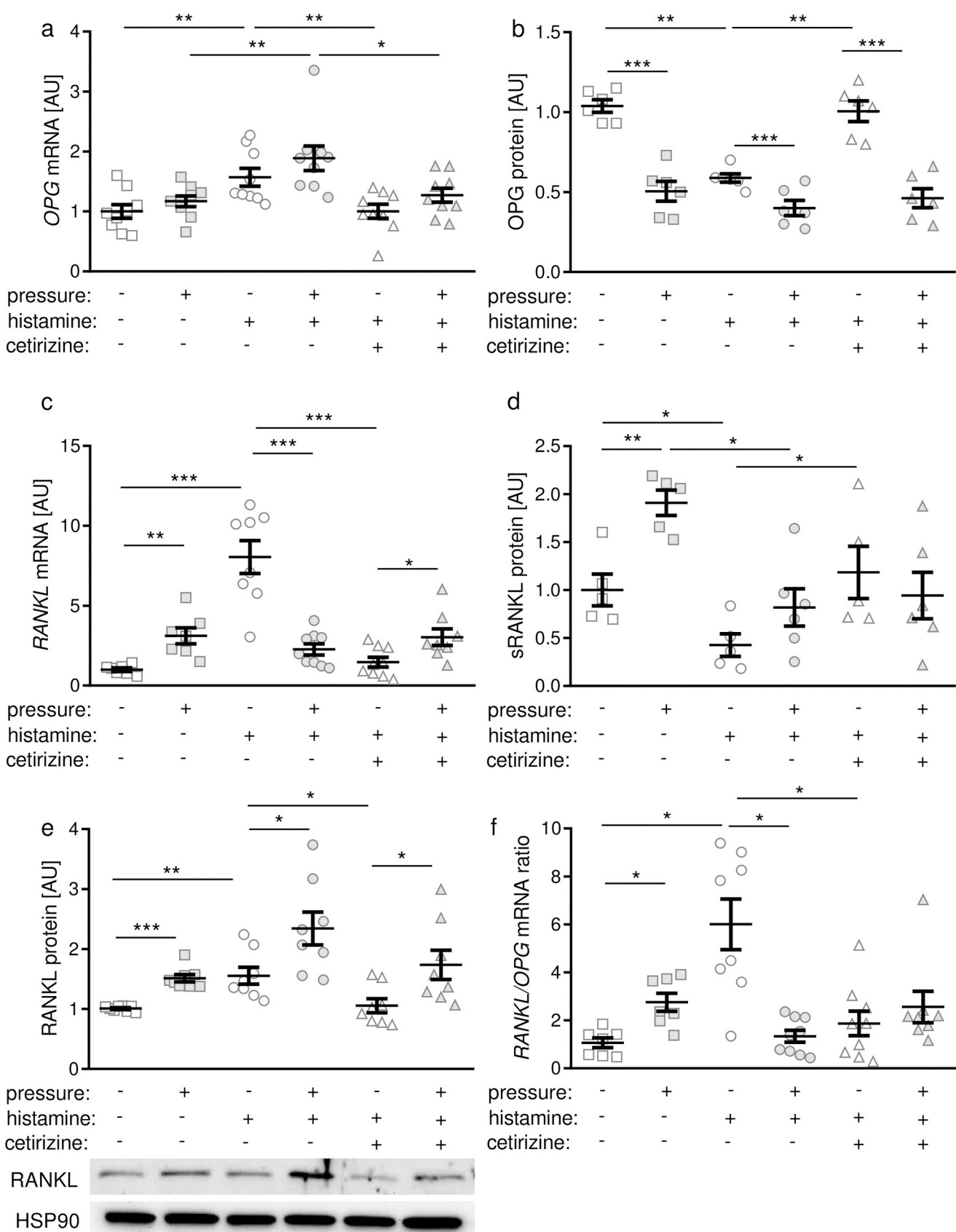

**Fig 5.** Effects of histamine and H1R antagonist cetirizine on OPG **(a,b)** and RANKL **(c-e)** gene and protein expression as well as on *RANKL/OPG* mRNA ratio **(f).** AU: arbitrary units; $^*$p $\leq$ 0.05; $^{**}$ p $\leq$ 0.01; $^{***}$ p $\leq$ 0.001. Statistics: Welch-corrected ANOVA with Games-Howell posthoc tests. Each symbol in figures represents a data point. Horizontal lines represent the mean ± standard error of mean.

diseases [39] than in orthodontic tooth movement. The highest receptor expression was observed regarding *H2R*, when adding 200 μg of histamine. At the same time we could not see any inhibiting effect when applying H2R-antagonist ranitidine. In addition, cetirizine showed the highest reversion of the histamine-induced effect. This was in line with results by Park et al. [32]. These authors investigated gene expression of nasal fibroblasts and determined *H1R* to be the most distinctly expressed receptor and a *H1R* antagonist having the highest inhibiting effect. Furthermore, our study indicates that histamine stimulates proliferation of PDLF. These results were in line with Hong et al., who reported an increasing cell number after applying histamine on nasal fibroblasts [40].

One of the first responses to orthodontic pressure is the synthesis of prostaglandins [41]. This is mediated by *COX-2*, an enzyme, which enhances inflammatory reactions [42]. As expected, gene expression of *COX-2* and secretion of PG-E2 increased in PDLF upon histamine and pressure treatment indicating an enhanced proinflammatory response at the beginning of orthodontic tooth movement. These data were in line with the study of Grimm et al., who reported a significant upregulation of *COX-2* and *IL-6* gene expression in PDLF within three hours [43] and with Niisato et al, who reported increased PG-E2 secretion with histamine treatment [29]. Studies with a similar setup showed an increased *COX-2*-induced PG-E2 expression during compressive force treatment [6,44,45]. Other studies explained the important role of PG-E2 for bone resorption [46] and its impact on RANKL expression [47].

Interleukin 6 (IL-6) plays an important role in host defense [9] and has an effect on bone resorption [48]. Histamine seems to have a decreasing effect on IL-6 expression during force treatment, which is less pronounced without pressure. Schroeder et al. reported an increase of *IL-6* expression during the first 48 h under orthodontic compressive forces and a decrease after 72 hours [6]. It is known that IL-6 is also regulated by IL1α/β and TNFα [44]. Okada et al. also reported that IL1α/β- or TNFα- induced IL-6 production can be inhibited by PG-E2. This could be a reason for the decrease of IL-6 expression after histamine treatment. Histamine enhanced expression of proinflammatory cytokines. Meh et al. (2011) found a correlation between tooth movement, histamine and cetirizine in rats. Tooth movement was increased by histamine and inhibited by cetirizine in the last period of orthodontic tooth movement [49]. In contrast, Kriznar et al. (2008) observed that cetirizine inhibited tooth movement in the first period of orthodontic tooth movement [50].

RANKL and its decoy receptor osteoprotegerin play an important role in bone formation and resorption [51]. RANKL binds to its receptor RANK on osteoclast precursor cells to stimulate osteoclast formation and activation [9]. Schroeder et al. found an increasing effect on RANKL expression during the first 72 h of orthodontic force treatment [6]. Nam et al. reported an upregulation of RANKL in the serum and nasal mucosal tissue of allergic rhinitis patients [52]. This is in line with our data, as RANKL gene expression and protein expression are significantly increased when adding histamine.

Based on these *in vitro* results, it is likely that increased histamine levels as occuring in patients with allergies may boost orthodontic tooth movement velocity, which is a sterile pseudo-inflammatory reaction dependent on an increase in inflammatory cytokines and RANKL expression leading to elevated osteoclastogenesis in direction of movement [3]. On the other hand, it is also possible that the elevated release of proinflammatory cytokines and RANKL by PDLF under the influence of histamine may trigger uncontrolled osteoclastogenesis

leading to severe side effects such as dental root resorptions and periodontal bone loss, which merits investigation in further *in vivo* studies.

## Conclusions

By expressing *H1R*, *H2R* and *H4R*, PDLF are likely to be able to detect fluctuating histamine levels in the periodontal ligament. Increased histamine levels seem to be associated with increased expression of proinflammatory mediators and RANKL, suggesting an inductive effect of histamine on PDLF-mediated osteoclastogenesis and thus orthodontic tooth movement, which requires resorption of the alveolar bone in direction of movement, but may also be associated with side effects such as dental root resorptions or periodontal bone loss during orthodontic therapy, which are caused by increased and uncontolled osteoclast activity. Since cetirizine as specific H1R inhibitor cancels these effects, the histamine effect seems to be predominantly mediated via the H1R.

## Supporting information

**S1 Fig. Fold changes in COX-2 and RANKL gene expression by PDLF due to pressure application for 48 h for each individual subject included into the used pool of PDLF.**
(DOCX)

**S2 Fig.  Effects of histamine and 50 μM H1R antagonist fexofenadine (F9427, Sigma-Aldrich) on *COX-2* (a) and *IL-6* (b) gene expression.** AU: arbitrary units; $^*$p $\leq$ 0.05; $^{**}$ p $\leq$ 0.01. Statistics: Welch-corrected ANOVA with Games-Howell posthoc tests. Each symbol in figures represents a data point. Horizontal lines represent the mean ± standard error of mean.
(DOCX)

**S1 Dataset.**
(XLSX)

**S1 Raw Images.**
(DOCX)

## Acknowledgments

The authors thank Kathrin Bauer and Eva Zaglauer for their technical support.

## Author Contributions

**Conceptualization:** Marcella Groeger, Gerrit Spanier, Michael Wolf, James Deschner, Peter Proff, Agnes Schröder, Christian Kirschneck.

**Data curation:** Marcella Groeger.

**Formal analysis:** Agnes Schröder, Christian Kirschneck.

**Funding acquisition:** Agnes Schröder, Christian Kirschneck.

**Investigation:** Marcella Groeger, Agnes Schröder.

**Methodology:** Marcella Groeger, James Deschner, Agnes Schröder, Christian Kirschneck.

**Project administration:** Peter Proff, Christian Kirschneck.

**Resources:** Gerrit Spanier.

**Software:** Agnes Schröder, Christian Kirschneck.

**Supervision:** Michael Wolf, James Deschner, Peter Proff, Agnes Schröder, Christian Kirschneck.

**Validation:** Gerrit Spanier, Michael Wolf, James Deschner, Peter Proff, Agnes Schröder, Christian Kirschneck.

**Visualization:** Agnes Schröder, Christian Kirschneck.

**Writing – original draft:** Marcella Groeger, Agnes Schröder, Christian Kirschneck.

**Writing – review & editing:** Marcella Groeger, Gerrit Spanier, Michael Wolf, James Deschner, Peter Proff, Agnes Schröder, Christian Kirschneck.

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
