## [Decision Letter · Decision Letter 0]

12 Jun 2020

PONE-D-20-13071

Effects of histamine on periodontal ligament fibroblasts during simulated orthodontic compressive strain

PLOS ONE

Dear Dr. Kirschneck,

Thank you for submitting your manuscript to PLOS ONE. After careful consideration, we feel that it has merit but does not fully meet PLOS ONE’s publication criteria as it currently stands. Therefore, we invite you to submit a revised version of the manuscript that addresses the reviewers' requests for minor modifications.

We look forward to receiving your revised manuscript.

Kind regards,

David M. Ojcius

Academic Editor

PLOS ONE

Journal Requirements:

Reviewers' comments:

Reviewer's Responses to Questions

**Comments to the Author**

1. Is the manuscript technically sound, and do the data support the conclusions?

Reviewer #1: Yes

Reviewer #2: Yes

2. Has the statistical analysis been performed appropriately and rigorously? 

Reviewer #1: Yes

Reviewer #2: Yes

3. Have the authors made all data underlying the findings in their manuscript fully available?

Reviewer #1: Yes

Reviewer #2: Yes

4. Is the manuscript presented in an intelligible fashion and written in standard English?

Reviewer #1: Yes

Reviewer #2: Yes

5. Review Comments to the Author

Reviewer #1: This study investigates the effects of histamine on PDLFs with or without orthodontic compression forces. Up to now, few studies were focused on histamine in periodontal areas regarding its release, concentration, and function, not to mention detecting its influence of PDLFs with active forces. This study's results fill this research gap and was performed in a generally comprehensive and logical way.

1. All target biomarkers were assessed in transcription and translation levels, which helped to further verify the obtained results.

2. Histamine receptor antagonists were adopted to rescue the effects of histamine. This also made the conclusion quite convincing.

Some minor revisions are suggested.

1. The figures were not concise and not clear. It is recommended to reduce the size of individual data points.

2. The ratio of RANKL/OPG should be described or demonstrated in the article.

3. The individual variation of PDLFs characteristics should be discussed as it is a pooled sample.

4. There are some typos in the literature and the conclusion seems incomplete.

Reviewer #2: Allergic reaction commonly exists in the population, including orthodontic patients. However, the effects of histamine, the critical mediator of allergy on orthodontic tooth movement, are not clearly understood. In this manuscript, the authors studied the effects of histamine on human periodontal ligament fibroblasts using an in vitro orthodontic compression cell culture model. As found, PDL fibroblasts expressed histamine receptors HIR, H2R, and H4R, but not H3R. Histamine increased the expressions of these receptors as well as IL-6 and COX2 and PGE2 release with and without compression. Histamine also increased the expression of RANKL, indicating its direct role in mediating osteoclastogenesis. These enhancements were almost abolished by H1R antagonist cetirizine. These findings add new knowledge to our understanding of the influence of allergic reactions in orthodontic tooth movement. Overall this is an interesting study but needs a major revision to be considered for publication.

Comments are listed below.

The title is better to be changed to “Effects of histamine on human periodontal ligament fibroblasts under simulated orthodontic pressure.”

The word “strain” in this manuscript should be replaced by “stress” or “pressure” or “compression”.

How specific is the H1R antagonist cetirizine? Any cross-reaction with other Histamine receptors e.g., H1R or H4R? How was the concentration of cetirizine determined? Any reference?

Please clarify the order to adding drugs. In other words, was the antagonist cetirizine added 2 hours before the addition of histamine, then followed by 48 hours with or without compress? If so, why this order was chosen?

The article “Hwang S, Chung CJ, Choi YJ, Kim T, Kim KH. The effect of cetirizine, a histamine 1 receptor antagonist, on bone remodeling after calvarial suture expansion. Korean J Orthod. 2020 Jan;50(1):42-51.” should be cited.

More discussion should be made on the relationship between the findings with orthodontic tooth movement. For example, will tooth movement be faster in patients with an allergy? Will orthodontically-induced root resorption and periodontal bone loss be more severe in patients with an allergy?

Are the concentrations of exogenous (added) histamine related to the histamine levels found in the patients with an allergy? Especially the levels of histamine in the PDL?

There is no legend of Figure 1.

The symbols representing data points are not necessary for all the Figures 2-5. Please remove them and make the Figures straight forward and clear.

Can the western blot in Figure 5 be replaced with a better one?

On line 416, the sentence is incomplete.

6. PLOS authors have the option to publish the peer review history of their article (what does this mean?). If published, this will include your full peer review and any attached files.

Reviewer #1: No

Reviewer #2: No

---

## [Author Response · Author response to Decision Letter 0]

13 Jul 2020

Reviewer #1

This study investigates the effects of histamine on PDLFs with or without orthodontic compression forces. Up to now, few studies were focused on histamine in periodontal areas regarding its release, concentration, and function, not to mention detecting its influence of PDLFs with active forces. This study's results fill this research gap and was performed in a generally comprehensive and logical way.

1. All target biomarkers were assessed in transcription and translation levels, which helped to further verify the obtained results.

2. Histamine receptor antagonists were adopted to rescue the effects of histamine. This also made the conclusion quite convincing.

Some minor revisions are suggested.

1. The figures were not concise and not clear. It is recommended to reduce the size of individual data points.

We reduced size of the individual data points to enhance clarity of the figures, following reviewer´s suggestion.

2. The ratio of RANKL/OPG should be described or demonstrated in the article.

According to reviewer’s suggestion, we calculated and added the RANKL/OPG mRNA ratio in Fig. 5. 

Next, we calculated RANKL/OPG mRNA ratio to directly assess the changes due to histamine or cetirizine treatment (Fig. 5f). Under control conditions compression resulted in an increased RANKL/OPG mRNA ratio. Without pressure histamine elevated RANKL/OPG mRNA ratio significantly mediated by the H1R, as this effect was truncated by cetirizine (Fig. 5f).

Fig 5. Effects of histamine and H1R antagonist cetirizine on OPG (a,b) and RANKL (c-e) gene and protein expression as well as on RANKL/OPG mRNA ratio (f). AU: arbitrary units; *p ≤ 0.05; ** p ≤ 0.01; *** p ≤ 0.001. Statistics: Welch-corrected ANOVA with Games-Howell posthoc tests. Each symbol in figures represents a data point. Horizontal lines represent the mean ± standard error of mean.

3. The individual variation of PDLFs characteristics should be discussed, as it is a pooled sample. 

We tested PDLF from each individual subject prior to generating the PDLF pool to assess individual variances in the reaction to compressive force stimuli. All of them reacted with enhanced COX-2 (ranging from 1.3 to 2.3) and RANKL (ranking from 1.5 to 4.3) gene expression compared to untreated controls. We included this information as Supplemental Figure 1.

PDLF cells of each individual subject included into the pool were tested for increased COX-2 and RANKL gene expression upon compressive force treatment (Supplemental Figure 1). 

Despite some interindividual variation of PDLF characteristics and expression behaviour, our results derived from a pool of PDLF from six gender-matched patients should be generalisable, as comparable upregulating effects by compressive force treatment were observed in all individual cell lines.

4. There are some typos in the literature and the conclusion seems incomplete.

Typos were corrected and the incomplete sentence in the conclusions section completed.

By expressing H1R, H2R and H4R, PDLF are likely to be able to detect fluctuating histamine levels in the periodontal ligament. Increased histamine levels seem to be associated with increased expression of proinflammatory mediators and RANKL, suggesting an inductive effect of histamine on PDLF-mediated osteoclastogenesis and thus orthodontic tooth movement, which requires resorption of the alveolar bone in direction of movement, but may also be associated with side effects such as dental root resorptions or periodontal bone loss during orthodontic therapy, which are caused by increased and uncontolled osteoclast activity. Since cetirizine as specific H1R inhibitor cancels these effects, the histamine effect seems to be predominantly mediated via the H1R.

Reviewer #2

Allergic reaction commonly exists in the population, including orthodontic patients. However, the effects of histamine, the critical mediator of allergy on orthodontic tooth movement, are not clearly understood. In this manuscript, the authors studied the effects of histamine on human periodontal ligament fibroblasts using an in vitro orthodontic compression cell culture model. As found, PDL fibroblasts expressed histamine receptors HIR, H2R, and H4R, but not H3R. Histamine increased the expressions of these receptors as well as IL-6 and COX2 and PGE2 release with and without compression. Histamine also increased the expression of RANKL, indicating its direct role in mediating osteoclastogenesis. These enhancements were almost abolished by H1R antagonist cetirizine. These findings add new knowledge to our understanding of the influence of allergic reactions in orthodontic tooth movement. Overall this is an interesting study but needs a major revision to be considered for publication.

Comments are listed below.

The title is better to be changed to “Effects of histamine on human periodontal ligament fibroblasts under simulated orthodontic pressure.”

Thank you for this suggestion. We changed the title accordingly.

The word “strain” in this manuscript should be replaced by “stress” or “pressure” or “compression”.

Thank you for that point. We replaced “strain” throughout the text and now use “pressure application” or “compression” following your suggestion.

How specific is the H1R antagonist cetirizine? Any cross-reaction with other Histamine receptors e.g., H1R or H4R? How was the concentration of cetirizine determined? Any reference?

To further support our data, we included experiments with fexofenadine into the supplementary information. Like cetirizine, fexofenadine also acts as H1 receptor antagonist. Comparable effects with fexofenadine were found on COX-2 and IL-6 gene expression as with cetirizine. Furthermore when taking a look into the literature, cetirizine has been shown to have an exquisite anti-H1 specificity: cetirizine appears unique in being devoid of action on receptors other than the H1 receptor (Bernheim et al. 1991). Regarding the used concentration of cetirizine, we adopted a concentration of a H1R antagonist previously used and published for experiments on nasal fibroblasts (Park et al. 2014).This information was added and respective literature cited in the manuscript. 

Cetirizine has been shown to have an exquisite anti-H1R specificity exerting its effects only on H1R (Bernheim et al. 1991) and to facilitate bone formation by suppressing osteoclastic activity (Hwang et al. 2020).

The used antagonist concentrations were adopted from the concentration of a H1R antagonist previously used and published in experiments on nasal fibroblasts (Park et al. 2014).

Please clarify the order to adding drugs. In other words, was the antagonist cetirizine added 2 hours before the addition of histamine, then followed by 48 hours with or without compress? If so, why this order was chosen?

According to reviewer´s suggestions we clarified that point in the Material and Method section. We preincubated the cells with cetirizine to block H1R before adding histamine. As we started with a preincubation with different histamine concentrations before adding pressure, we decided to keep this experimental setup for the HR antagonist experiments as well.

To test for histamine receptor (HR) interaction, we additionally incubated PDLF with 100 μM of H1R antagonist cetirizine (C3618, Sigma Aldrich, Munich, Germany), H2R antagonist ranitidine (R101, Sigma Aldrich, Munich, Germany) or H4R antagonist JNJ777120 (J3770, Sigma Aldrich, Munich, Germany), respectively, two hours prior to histamine application. Then PDLF were preincubated for 24 h followed by pressure application for another 48 h as described above. We then analyzed gene expression (RT-qPCR) and protein expression (Western Blot, ELISA).

The article “Hwang S, Chung CJ, Choi YJ, Kim T, Kim KH. The effect of cetirizine, a histamine 1 receptor antagonist, on bone remodeling after calvarial suture expansion. Korean J Orthod. 2020 Jan;50(1):42-51.” should be cited.

Following reviewer´s suggestion, we cited this study accordingly.

Cetirizine has been shown to have an exquisite anti-H1R specificity exerting its effects only on H1R (Bernheim et al. 1991) and to facilitate bone formation by suppressing osteoclastic activity (Hwang et al. 2020).

More discussion should be made on the relationship between the findings with orthodontic tooth movement. For example, will tooth movement be faster in patients with an allergy? Will orthodontically-induced root resorption and periodontal bone loss be more severe in patients with an allergy?

As suggested, we added a discussion about likely clinical effects of histamine on orthodontic tooth movement and side effects to expand the scope of the study for the general reader.

Based on these in vitro results, it is likely that increased histamine levels as occuring in patients with allergies may boost orthodontic tooth movement velocity, which is a sterile pseudo-inflammatory reaction dependent on an increase in inflammatory cytokines and RANKL expression leading to elevated osteoclastogenesis in direction of movement (Meikle 2006). On the other hand, it is also possible that the elevated release of proinflammatory cytokines and RANKL by PDLF under the influence of histamine may trigger uncontrolled osteoclastogenesis leading to severe side effects such as dental root resorptions and periodontal bone loss, which merits investigation in further in vivo studies.

Are the concentrations of exogenous (added) histamine related to the histamine levels found in the patients with an allergy? Especially the levels of histamine in the PDL?

Actually we found no data regarding histamine release in periodontal ligament during allergic reaction as this is difficult to measure experimentally. However, we tested different histamine concentrations (incrementally increasing from 50 to 100, 150 and 200 µM) and found 100 µM histamine have the most distinct effects on PDLF. This was in line with other studies also using 100 µM histamine for their experiments with PDLF (Niisato et al. 1996) or even 200 µM for nasal fibroblasts (Park et al. 2014). Based on our own results and the lower end of the concentration range found in the literature for fibroblasts (100-200 µM), we decided to use 100 µM to probably yield the most representative results. 

There is no legend of Figure 1.

Legend for Fig. 1 was integrated in the text after its first mentioning (line 140-145)

Fig 1. In vitro simulation of compressive force application to periodontal ligament fibroblasts (PDLF) occurring during orthodontic tooth movement. After a preincubation time of 24 h with or without histamine (100 μM, 50/100/200 μM in receptor expression experiments) and an H1R/H2R/H4R antagonist (100 μM), a pressure of 2g/cm2 was applied to PDLF by means of a sterile glass disc (ø33cm, 17.1g) for 48 h according to an established and published in vitro model.

The symbols representing data points are not necessary for all the Figures 2-5. Please remove them and make the Figures straightforward and clear.

We have decided to represent individual points as symbols in order to increase the informative value of the figures and to be transparent in our reporting by showing our raw data. It is clear to us that this will make the images more confusing. To alleviate this, we implemented reviewer's #1 suggestion and reduced the size of the symbols rather than deleting them completely and believe that this will make the figures more straightforward and clear.

Can the western blot in Figure 5 be replaced with a better one?

According to reviewer´s suggestion, we added a new western blot image in Fig. 5

On line 416, the sentence is incomplete.

The incomplete sentence in the conclusions section was completed.

By expressing H1R, H2R and H4R, PDLF are likely to be able to detect fluctuating histamine levels in the periodontal ligament. Increased histamine levels seem to be associated with increased expression of proinflammatory mediators and RANKL, suggesting an inductive effect of histamine on PDLF-mediated osteoclastogenesis and thus orthodontic tooth movement, which requires resorption of the alveolar bone in direction of movement, but may also be associated with side effects such as dental root resorptions or periodontal bone loss during orthodontic therapy, which are caused by increased and uncontolled osteoclast activity. Since cetirizine as specific H1R inhibitor cancels these effects, the histamine effect seems to be predominantly mediated via the H1R.

Bernheim, J.; Arendt, C.; Vos, C. de (1991): Cetirizine: more than an antihistamine? In: Agents Actions Suppl 34, S. 269–293.

Hwang, Soonshin; Chung, Chooryung J.; Choi, Yoon Jeong; Kim, Taeyeon; Kim, Kyung-Ho (2020): The effect of cetirizine, a histamine 1 receptor antagonist, on bone remodeling after calvarial suture expansion. In: Korean J Orthod 50 (1), S. 42–51. DOI: 10.4041/kjod.2020.50.1.42.

Meikle, Murray C. (2006): The tissue, cellular, and molecular regulation of orthodontic tooth movement: 100 years after Carl Sandstedt. In: Eur J Orthod 28, S. 221–240. DOI: 10.1093/ejo/cjl001.

Niisato, N.; Ogata, Y.; Furuyama, S.; Sugiya, H. (1996): Histamine H1 receptor-stimulated Ca2+ signaling pathway in human periodontal ligament cells. In: J Periodont Res 31, S. 113–119. DOI: 10.1111/j.1600-0765.1996.tb00472.x.

Park, Il-Ho; Um, Ji-Young; Cho, Jung-Sun; Lee, Seung Hoon; Lee, Sang Hag; Lee, Heung-Man (2014): Histamine Promotes the Release of Interleukin-6 via the H1R/p38 and NF-κB Pathways in Nasal Fibroblasts. In: Allergy Asthma Immunol Res 6, S. 567–572. DOI: 10.4168/aair.2014.6.6.567.

---

## [Decision Letter · Decision Letter 1]

20 Jul 2020

Effects of histamine on human periodontal ligament fibroblasts under simulated orthodontic pressure

PONE-D-20-13071R1

Dear Dr. Kirschneck,

We’re pleased to inform you that your manuscript has been judged scientifically suitable for publication and will be formally accepted for publication once it meets all outstanding technical requirements.

Kind regards,

David M. Ojcius

Academic Editor

PLOS ONE

Additional Editor Comments (optional):

Reviewers' comments:

Reviewer's Responses to Questions

**Comments to the Author**

1. If the authors have adequately addressed your comments raised in a previous round of review and you feel that this manuscript is now acceptable for publication, you may indicate that here to bypass the “Comments to the Author” section, enter your conflict of interest statement in the “Confidential to Editor” section, and submit your "Accept" recommendation.

Reviewer #1: All comments have been addressed

2. Is the manuscript technically sound, and do the data support the conclusions?

Reviewer #1: Yes

3. Has the statistical analysis been performed appropriately and rigorously? 

Reviewer #1: Yes

4. Have the authors made all data underlying the findings in their manuscript fully available?

Reviewer #1: Yes

5. Is the manuscript presented in an intelligible fashion and written in standard English?

Reviewer #1: Yes

6. Review Comments to the Author

Reviewer #1: The comments have been addressed with some more data added. The revision is satisfactory. The figures are more clear.

7. PLOS authors have the option to publish the peer review history of their article (what does this mean?). If published, this will include your full peer review and any attached files.

Reviewer #1: No